Global gene expression analyses of the alkamide-producing plant Heliopsis longipes supports a polyketide synthase-mediated biosynthesis pathway

http://orcid.org/0000-0001-8503-1372 Buitimea-Cantúa Génesis V. 1
Marsch-Martinez Nayelli 1
Ríos-Chavez Patricia 2
http://orcid.org/0000-0002-0991-5464 Méndez-Bravo Alfonso 3
http://orcid.org/0000-0003-2371-0307 Molina-Torres Jorge 1 jmolinat@cinvestav.mx
1 Department of Biotecnologia y Bioquímica, CINVESTAV Unidad Irapuato , Irapuato, Guanajuato , Mexico
2 Instituto de Investigaciones Químico-Biológicas, Universidad de San Nicolás de Hidalgo , Morelia, Michoacan , Mexico
3 Laboratorio Nacional de Análisis y Síntesis Ecológica, CONACYT – Escuela Nacional de Estudios Superiores , Morelia, Michoacan , Mexico
Lazo Gerard
Electronic publication date: 2020 Sep 25
Publication date: 2020
Volume: 8
Electronic Location ID: e10074
Received 2020 Feb 21; Accepted 2020 Sep 10
Copyright: © 2020 Buitimea-Cantúa et al.
Copyright year: 2020
Copyright holder: Buitimea-Cantúa et al.
License: This is an open access article distributed under the terms of the Creative Commons Attribution License, which permits unrestricted use, distribution, reproduction and adaptation in any medium and for any purpose provided that it is properly attributed. For attribution, the original author(s), title, publication source (PeerJ) and either DOI or URL of the article must be cited.
License URL: https://creativecommons.org/licenses/by/4.0/

Keywords: Affinin, Alkamides biosynthesis, Heliopsis longipes, Polyketide synthase, RNA sequencing, Transcriptome analysis, Plants PKS

Funding: Consejo Nacional de Ciencia y Tecnología (CONACYT, Mexico) Genesis V. Buitimea-Cantua’s work was supported by a doctoral scholarship from Consejo Nacional de Ciencia y Tecnología (CONACYT, Mexico). The funders had no role in study design, data collection and analysis, decision to publish, or preparation of the manuscript.

==============================
Background

Alkamides are plant-specific bioactive molecules. They are low molecular weight N-substituted α-unsaturated acyl amides that display biological explicit activities in different organisms from bacteria, fungi, insects to mammals and plants. The acyl chain has been proposed to be biosynthesized from a fatty acid; however, this has not been demonstrated yet. Heliopsis longipes (Asteraceae) accumulates in root a C10 alkamide called affinin in its roots, but not in leaves. The closely related species Heliopsis annua does not produce alkamides. To elucidate the biosynthetic pathway of the alkamides acyl chain, a comparative global gene expression analysis contrasting roots and leaves of both species was performed.

Methods

Transcriptomics analysis allowed to identify genes highly expressed in H. longipes roots, but not in tissues and species that do not accumulate alkamides. The first domain searched was the Ketosynthase (KS) domain. The phylogenetic analysis using sequences of the KS domain of FAS and PKS from different organisms, revealed that KS domains of the differentially expressed transcripts in H. longipes roots and the KS domain found in transcripts of Echinacea purpurea, another alkamides producer species, were grouped together with a high bootstrap value of 100%, sharing great similarity. Among the annotated transcripts, we found some coding for the enzymatic domains KS, AT, ACP, DH, OR and TE, which presented higher expression in H. longipes roots than in leaves. The expression level of these genes was further evaluated by qRT-PCR. All unigenes tested showed higher expression in H. longipes roots than in any the other samples. Based on this and considering that the acyl chain of affinin presents unsaturated bonds at even C numbers, we propose a new putative biosynthesis pathway mediated by a four modules polyketide synthase (PKS).

Results

The global gene expression analysis led to the selection of a set of candidate genes involved in the biosynthesis of the acyl chain of affinin, suggesting that it may be performed by a non-iterative, partially reductive, four module type I PKS complex (PKS alk) previously thought to be absent from the plant kingdom.

Introduction

Alkamides are plant natural products that display diverse biological activities in different phyla, such as antifungal, antibacterial, antimalarial, insecticidal, quorum sensing blockage, anesthetic, analgesic and pungent activities (Barbosa et al., 2016). They also contribute to plant growth promotion and gene defense induction and interact with intercellular signal molecules in humans, bacteria and plants (Ramírez-Chávez et al., 2004; Méndez-Bravo et al., 2011). Chemically, alkamides are low molecular weight, <400 Da, N-substituted α-unsaturated acyl amides. They consist of a straight α-unsaturated C8–C18 acyl chain condensed to different amines. The most abundant alkamide acyl chain reported is C10 (Greger, 2016). The acyl chain of these rather simple structures presents unsaturated double bonds found isolated (E) or conjugated (E, E or E, Z), either in the α-, intermediate or methyl-ω position, starting at an even C position. Alkamides are plant metabolites comprising a group close to 300 structures produced by a limited number of plant species belonging taxonomically to only eight families. A database of alkamides and other acyl-amides has been recently compiled (Boonen et al., 2012).

Alkamides are synthetized by plants, and similar, higher molecular weight ≳800 Da, onnamide-type polyketides are produced by sponge from symbiotic bacteria (Piel, 2009). In plants, alkamide accumulation is restricted to specific tissues such as roots, flowers, fruits or seeds, except in the genus Acmella (Asteraceae) where alkamides have been detected in the whole plant. Several species containing high levels of alkamides are found in the Asteraceae, Piperaceae and Rutaceae; being more frequent in the Asteraceae family (Christensen & Lam, 1991; Parmar et al., 1997; Yang, 2008). The Heliopsis genus, which belongs to the Heliantheae tribe in the latter family, contains 16 species, most of them endemic to Mesoamerica (García-Chávez, Ramírez-Chávez & Molina-Torres, 2004; Ramírez-Noya, González-Elizondo & Molina-Torres, 2011). Plants of this genus produce over 30 different alkamides. Their acyl chain length varies from C10 to C18, with the amine moiety being usually either isobutylamide or 2-methylbutylamide. The most commonly reported functional groups of plant alkamides are isobutylamine as the amine moiety and the C10 acyl chain. The α-double bond of the acyl moiety is the main characteristic of alkamides. In several genus for example, Heliopsis, other double bonds vary in position but are frequently conjugated at positions 4, 6, 8, 10 or 14. Triple (acetylenic) bonds, when present, are in a position corresponding to that of the double bonds, most frequently occurring as acetylenic or methyl acetylenic in the ω-position. When more than one acetylenic bond is present, these are conjugated. In those alkamides where the acetylenic bond is at the ω-position, the acyl chain presents an odd number of carbons. In Heliopsis longipes, alkamides accumulate in the roots but are not present in the vegetative aerial tissue. In contrast, alkamides have not been detected in leaves or roots of the related species Heliopsis annua. The most abundant alkamide in H. longipes roots is N-isobutyl-2E,6Z,8E-decatrienamide, named affinin. It represents over 90% of the total alkamide content in well-developed roots. Other C10 acyl chain alkamides in this tissue are structurally related to affinin, such as the N-2-methylbutyl-2E,6Z,8E-decatrienamide and the bornyl ester of C10-2E,6Z,8E (Fig. 1). They contain the same acyl chain, C10-2E,6Z,8E, and therefore these alkamides might be produced in a similar biosynthetic pathway not yet completely elucidated. Information is available regarding the biosynthetic pathway of the amine moiety. Studies performed on some species of the Asteraceae have suggested that the amine moiety isobutyl of affinin is derived from an amino acid that is first decarboxylated and then linked through an amide bond to the acyl chain. This was confirmed through feeding experiments using L-[2H8]-valine, employing Acmella radicans as a model (Cortez-Espinosa et al., 2011). Moreover, a recent detailed study on alkamides biosynthesis in several Echinacea species using transcriptomics, metabolite profiling and isotopic labeling, validated that a branched-chain amine is acylated with the fully extended acyl chain. A pyridoxal phosphate-dependent decarboxylase was identified as the enzyme responsible for generating the amide alkamide moiety (Rizhsky et al., 2016).

Figure 1 General structure of alkamides (alk) and the main alkamides produced in Heliopsis longipes roots.

(A) R1: α-unsaturated acyl chain C8–C18. Other double bonds may be present isolated (E) or conjugated (E,E or E,Z). R2: amine. R3: mostly H. (B) Structure of affinin (N-isobutyl-2E,6Z,8E-decatrienamide) the most abundant, represents over 90% of the total alkamides in roots. Minor alkamides are: (C) N-2-methylbutyl-2E,6Z,8E-decatrienamide (3.3%), (D) N-isobutyl-2E-en-8,10-diyn-undecanamide (2.5%), and (E) decanoic-2E,6Z,8E-bornyl-ester (1.1%). The latter, although not an alkamide, presents the same acyl moiety as the denoted alkamides.

The acyl chain of alkamide molecules has been considered to arise from a polyunsaturated fatty acid. This alkamide biosynthesis pathway mediated by fatty acids was proposed several decades ago based on feeding experiments with 14C- and 3H-labelled acetate precursors, demonstrating that these derived from acetate. More recently, a proposed path suggested that the alkamide acyl chain was mediated by oleic acid (C18:1) by successive β-oxidation, dehydration, and isomerization (Greger, 1984; Minto & Blacklock, 2008). However, this biosynthetic pathway for alkamides has not been experimentally confirmed. Fatty acids are distributed among all living organisms, indicating that all organisms possess a functional Fatty Acid Synthase (FAS) pathway, while alkamides are only present in some plant species. Structurally the double bonds of the acyl chain of alkamides are clearly different from those of fatty acids and their derivatives, which are always in the cis configuration (Z) and non-conjugated. The double bonds of fatty acids are incorporated by desaturases at a site different from that on which the acyl chain is synthesized and are strictly positioned in relation to the first desaturation 9Z, such as 12Z, 15Z, or 18Z (Brown, Slabas & Rafferty, 2009). In contrast, the acyl chains of alkamides contain double bonds in different positions of the chain. These double bonds are largely present in the trans configuration (E) but may be in cis (Z) when these are conjugated, in contrast to those found in fatty acids. These facts, together, suggest that the biosynthesis of the acyl chain of alkamides could occur through an alternative pathway. Biochemical or genetic information to support the fatty acid or any alternative biosynthetic pathway of alkamides is still scarce, essentially because alkamides are produced in non-model plants for which the genomes have not yet been sequenced. Whole genome sequencing has been successfully used to elucidate biosynthetic pathways in microorganisms, such as bacteria. However, plant genomes are much larger than bacterial genomes, which increases the cost and time associated with sequencing, and plant genomes exhibit particularities such as alternative splicing that can give rise to different transcript versions of a single gene. Therefore, transcript profiling using high-throughput RNA sequencing (RNA-Seq) is a first-rate alternative to investigate the biosynthetic pathways of metabolites in plants without sequenced genomes (Medema & Osbourn, 2016). Transcriptomic or global expression analyses can be a powerful tool, and comparison of cell or tissue transcriptomes that do or do not produce a metabolite of interest, can allow the discovery of the genes involved in specific biosynthetic pathways. Transcriptome sequencing by RNA-Seq has been successfully used to profile the expression patterns from different plant tissues, facilitating the acquisition of additional knowledge on secondary metabolite biosynthesis in plants, especially from non-model plant species (Osbourn & Lanzotti, 2009).

In this study we performed global expression analysis by RNA-Seq and compared the transcriptomes of tissues and species that accumulate or do not accumulate alkamides. Differentially expressed candidate genes were identified and based on their nature, an alternative putative biosynthesis pathway for the acyl chain of alkamides is proposed. The results here obtained provide a robust basis for subsequent informed experiments to confirm and further understand the proposed biosynthesis pathway of alkamides, specifically for the acyl chain of these molecules.

Materials and Methods

Biological material

Heliopsis longipes was collected in February 2014 at Puerto de Tablas, Xichú municipality, Sierra Gorda, state of Guanajuato, México (Lat. 21°14′20″N, Long. 100°05′19″W, Alt. 2,589 m above sea level). Voucher specimens were deposited at Instituto de Ecología, Centro Regional del Bajío. Herbarium IEB (Voucher 263787). Heliopsis annua was collected in October 2014 in the municipality of Vicente Guerrero, state of Durango, México (Lat. 23°44′41″N, Long. 104°00′02″W, Alt. 1,950 m above sea level). Voucher specimens were deposited at CIIDIR Durango Herbarium (Voucher id. D. Ramírez 5543). Fresh roots and leaves of both species were collected separately, washed with distilled water, immediately frozen in liquid nitrogen, and stored at −80 °C until processed for RNA extraction.

RNA extraction and transcriptome sequencing

Total RNA was extracted from roots and leaves of eight individual plants of each Heliopsis species and processed separately as two replicates, following the protocol established by the PureLinkTM Micro-to-Midi Total RNA Purification System (Invitrogen, Carlsbad, CA, USA). The quality of the RNA samples was evaluated using the RNA 6000 Nanochip on the BioAnalyzer 2100 (Agilent Technologies, Cedar Creek, TX, USA). Complementary DNA libraries were constructed with insert sizes from 339 to 697 base-pairs (bp) and sequenced using the Illumina HiSeq 2500 platform (Illumina, San Diego, CA, USA) with paired-end reads of 100 bp, at the Unidad de Servicios Genómicos of CINVESTAV, Irapuato, Guanajuato, México.

De novo transcriptome assembly

The resulting FASTQ files were pre-processed by removing adapter sequences and low-quality bases using the software package Trimmomatic-0.32, as previously described (Bolger, Lohse & Usadel, 2014; Kamitani et al., 2016). Clean reads were assembled into unigenes using the software package Trinity 2.0 with the default parameters (Garber et al., 2011). To reduce redundancy, unigene clustering was performed using the software package CD-HIT 4.0 with the 95% identity parameter to generate a set of non-redundant contig sequence files (Fu et al., 2012). To estimate expression abundance, reads were mapped to the de novo transcriptome assemblies using Bowtie 2.0 (Langmead et al., 2009) and expression abundance was calculated using the RNAseq by Expectation-Maximization (RSEM) software (Li & Dewey, 2011).

Gene function annotation

The consensus contig sequence files were annotated against the non-redundant protein (NR) database from the National Center for Biotechnology Information (NCBI; www.ncbi.nlm.nih.gov), Protein Family (PFAM; https://pfam.xfam.org) database, which contains a large collection of protein families, each represented by multiple sequence alignments and hidden Markov models, and Swiss-Prot database (https://www.uniprot.org/), with an e-value cut-off of 1e−5. Gene Ontology (GO) analyses were further used to categorize the function of the unigenes with Blast2GO 5 (https://www.blast2go.com/). Using the basic local alignment search tool (BLAST), the Arabidopsis thaliana transcription factor (TF) database was queried to identify TFs among all unigenes (identity N80%) (Guo et al., 2005).

Differential expression analysis

To estimate the expression level of each transcript in the different tissue samples, high-quality reads from each sample were mapped onto the final transcriptome assembly using the RSEM software (Li & Dewey, 2011). Read counts were normalized by calculating fragments per kilobase of exon model per million mapped reads (FPKM) values. Differential gene expression analysis was performed using the EdgeR (Bioconductor, R) software (Robinson, McCarthy & Smyth, 2010). A p-value cut-off ≤0.05 [multiple test correction: false discovery rate (FDR)] and log2 fold change (FC) ≥2 were used to identify significant differences in transcript expression.

Phylogenetic analysis of KS domains

The amino acid sequences of the KS domains of H. longipes and H. annua were obtained from the translated assembled RNA-Seq data. The KS domains were corroborated by BLAST (https://blast.ncbi.nlm.nih.gov/Blast.cgi) and PFAM, a database of large collection with protein families (https://pfam.xfam.org). The domains compared were: the KS domains identified in H. annua roots (A1–A4); the KS domains upregulated in H. longipes roots (B1–B4); the KS domains related to FAS from H. annua (C1–C11); the KS related to FAS from H. longipes (D1–D11); the KS of the type III PKS from H. annua (E1–E4); and the KS of the type III PKS from H. longipes (F1–F4). These KS sequences were compared with KS domains of several microorganisms, plant and mammal species obtained from the NCBI database (www.ncbi.nlm.nih.gov) (Table S1). The KS sequence of Echinacea purpurea was retrieved from the medicinal plant database of the Plant Biology Department, Michigan State University, East Lansing, MI, USA (http://medicinalplantgenomics.msu.edu/). Echinacea purpurea also belongs to tribe Heliantheae within the Asteraceae and produces alkamides that are structurally related to affinin (Rizhsky et al., 2016). After selection, sequences were aligned using ClustalW with the default parameters (Chenna, 2003). Phylogenetic and molecular evolutionary analyses were conducted in MEGA 7 using the maximum likelihood method based on the JTT matrix-based model and 1,000 bootstrap replications (Jones, Taylor & Thornton, 1992; Kumar, Stecher & Tamura, 2016).

Quantitative RT-PCR

Reverse Transcriptase quantitative polymerase chain reaction assays (RT-qPCR) were performed using the Applied Biosystems 7500 Fast Real-Time PCR system (Applied Biosystems, Foster City, CA, USA) with the Brilliant II SYBR Green RT-qPCR 1-Step Master Mix Kit (Agilent Technologies, Cedar Creek, TX, USA). Gene-specific primers for the selected genes in the acyl chain biosynthesis pathway were designed using Primer3 (www.primer3.org) and are shown in Table S2. The RT-qPCR for each gene was performed using three biological and three technical replicates. Actin served as the internal control for comparison. Relative transcript expression levels were calculated using the 2−ΔΔCT method (Chenna, 2003; Livak & Schmittgen, 2001; Bustin et al., 2009). Differential gene expression analysis based on the RT-qPCR results was performed for H. longipes roots vs. leaves and H. longipes roots vs. H. annua roots.

Results

RNA sequencing and de novo transcriptome assembly

To find differentially expressed genes that could encode enzymes involved in alkamide biosynthesis, transcriptomes of producing and non-producing tissues and species were obtained. Roots of H. longipes produce alkamides, while leaves do not. Neither roots nor leaves of H. annua produce these compounds. RNA was isolated from each tissue and RNA-Seq was performed. Each library was sequenced using the Illumina Hiseq 2500 platform. Adapter sequences, low quality reads, and reads that were shorter than 50 bp were removed, yielding more than 275,538,680 clean reads in the 2 × 100 paired-end formats (c. 27.7 Gbp). Roots and leaves of H. longipes generated 45,622,318 and 31,731,475 raw reads, respectively, whereas roots and leaves of H. annua generated 30,326,119 and 30,089,428 raw reads, respectively (Table 1A). Clean reads were assembled de novo using the Trinity package, developed specifically for next-generation short-read sequences (Garber et al., 2011). Based on the high-quality reads, 165,770 unigenes in H. longipes and 155,457 unigenes in H. annua were assembled (Table 1B). In terms of size distribution: 70% of the H. longipes unigenes and 77% of the H. annua unigenes were between 200 and 1,000 bp. Transcripts in H. longipes had a maximum size of 14,461 bp, an average size of 906 bp, and a minimum size of 200 bp. In H. annua, transcripts had a maximum size of 15,574 bp, an average size of 695 bp, and a minimum size of 200 bp.

Table 1 RNA sequencing (RNA-Seq) and de novo assembly data for the roots and leaves transcriptomes.

Sample	cDNA library (bp)	Read 1	Read 2	Run mode	Total
(GB)	
(A) RNA-Seq						
Heliopsis longipes roots 1	417	22,249,016	22,249,016	2 × 100 bp	27.7	
Heliopsis longipes roots 2	854	23,373,302	23,373,302	
Heliopsis longipes leaves 1	339	16,065,446	16,065,446	
Heliopsis longipes leaves 2	697	15,666,029	15,666,029	
Heliopsis annua roots 1	448	15,518,097	15,518,097	
Heliopsis annua roots 2	474	14,808,022	14,808,022	
Heliopsis annua leaves 1	497	15,754,789	15,754,789	
Heliopsis annua leaves 2	483	14,334,639	14,334,639	
(B) de novo assembly	
	Heliopsis longipes	Heliopsis annua	
Bases in unigenes	150,124,843	115,219,511	
Total unigenes	165,770	155,457	
Maximum size	14,461	15,574	
Average size	906	695	
Minimum size	200	200	

Functional annotation

Functional annotations of the transcriptomes revealed functional predictions for 81,682 transcripts for H. longipes and 68,166 transcripts for H. annua. The sequences that did not present a significant hit with a known protein were most likely short sequences with less than 200 bp, and probably represented genes that have not yet been functionally characterized, untranslated regions or specific genes for Heliopsis species. Based on the top-hit transcript homologies against the NR database, ca. 90% of the unigenes in both species were highly homologous to proteins from land plants. The Gene Ontology classification system defines concepts/classes used to describe gene function and relationships between these concepts (Ashburner et al., 2000). Using this system, we were able to assign 29,593 GO terms to the unigenes. Annotations were distributed into the three major categories of GO: Molecular Function, molecular activities of gene products; Cellular Component, where gene products are active; and Biological Process, pathways and larger processes comprising the activities of multiple gene products. GO annotations included: for H. longipes 10,796 molecular functions, 14,601 biological processes, and 4,153 cellular components; for H. annua 11,018 molecular functions, 14,367 biological processes and 4,251 cellular components (Fig. 2). The distribution of unigenes into the three different GO categories is an indicator of the wide diversity of genes that were present in the roots and leaves transcriptomes of these species. The most commonly assigned functional categories in each species, “catalytic activity” for the molecular function category, and “metabolic process” for the biological process category, were consistent with the results of other non-model plant species producing specialized metabolites (Xiao et al., 2013). As in transcriptomes of other plants, some transcripts encoded proteins of unknown function.

Figure 2 Gene Ontology (GO) classification of unigenes derived from RNA-Seq in Heliopsis species studied.

A total of 29,316 unigenes in H. longipes and 29,870 unigenes in H. annua were successfully annotated and classified into three GO categories: cellular component, biological process and molecular function.

Transcription factors play key regulatory roles in the secondary metabolism of plants. The identification, annotation, and classification of TFs in the roots and leaves transcriptomes of H. longipes and H. annua revealed high homology with 42 known TF families. Among the 1,006 unigenes identified as TFs in H. longipes, the most abundant presented high homology to members of the WRKY family (241 unigenes), followed by the MYB family (223 unigenes), and members of the GRAS family (80 unigenes). In H. annua, among the 938 unigenes related to TFs, the most abundant presented high homology to members of the LSD family (218 unigenes), followed by the WRKY family (162 unigenes), and the members of the HB family (68 unigenes). Finding MYB family members in H. longipes was interesting, since the MYB TFs include a conserved MYB DNA-binding domain that commonly binds to and regulates genes that code for enzymes involved in metabolic pathways (Ambawat et al., 2013).

Differential gene expression analysis

The comparison of transcriptomes obtained from roots and leaves of H. longipes and H. annua revealed differences in transcript expression among the two tissues and Heliopsis species. Among the 5,818 differentially expressed transcripts identified in H. annua roots, compared to H. longipes roots, 4,997 transcripts were down-regulated and 821 were up-regulated. The comparison between transcriptomes of H. longipes roots and leaves revealed 2,058 differentially expressed transcripts: 1,399 were down-regulated and 659 were up-regulated in roots. For both Heliopsis species, the up-regulated transcripts in roots, when compared to their corresponding leaves, encoded some proteins of unknown function, whereas the down-regulated transcripts included proteins related to photosynthesis, carbohydrate metabolism, and other processes. The differential expression analyses also identified 71 genes related to the FAS complex in the two Heliopsis species studied. These genes were highly expressed in H. annua leaves. The differential expression analysis highlighted genes encoding enzymes that could be responsible for the biosynthesis of the acyl chain in the structure of alkamides, as represented by the C10-2E,6Z,8E acyl chain of affinin in H. longipes. When comparing the transcriptomes from both species with differential alkamide production and accumulation, we identified a set of genes whose expression was higher in H. longipes roots, representing the species and tissue that accumulated the metabolites of interest in the present study (Fig. 3A).

Figure 3 Relative expression of candidate genes involved in the putative acyl-chain biosynthetic pathway identified by differential gene expression analysis of Heliopsis longipes and Heliopsis annua roots and leaves transcriptomes.

(A) Quantitative real-time PCR. (B) Expression, in fragments per kilobase of exon model per million mapped reads (FPKM) from the RNA sequencing data. ACP, acyl carrier protein; AT, acyl transferase; DH, dehydratase; KS, ketoacyl synthase; OR, oxide reductase including the K: keto reductase; ER, enoyl reductase; DH, dehydratase; TE, thioesterase. The number corresponds to the transcript set identified by differential expression analysis. Error bars represent standard error for n = 2. The relative expression pattern of the candidate transcripts in the non-alkamide producing tissue was close to zero and therefore is not easily observed in the graphs.

In order to confirm the differential expression of candidate genes in H. longipes and H. annua leaves and roots an analysis by RT-qPCR was performed. Transcripts encoding enzymatic domains that could be involved in the biosynthesis of a molecule with the characteristics of alkamides were sought, choosing those involved in the metabolism of acetate. As such candidate genes encoding the different domains: ketosynthase (KS), acyl transferase (AT), acyl carrier protein (ACP), oxide reductase (OR), dehydratase (DH) and thioesterase (TE) were selected (Table S2). Actin (ACT), coding a cytoskeleton structural protein, was used as a reference gene (Drobak, Franklin-Tong & Staiger, 2004), and the expression analyses were performed based on 2−ΔΔCT method (Chenna, 2003). The RT-qPCR data confirmed that the candidate genes were expressed, and that they showed differential expression, being highly expressed in H. longipes roots in comparison to H. longipes leaves and H. annua leaves and roots (Fig. 3B). RNA-Seq data revealed a good concordance with RT-qPCR expression quantifications. This has also been reported by other authors (Wang et al., 2016; Zhang & Deyholos, 2016). Our results also showed that all expression comparisons of the RT-qPCR assays were in fairly good match with the RNA-Seq data, even if the fold-change of some genes in their expression level detected by RNA-Sequencing and RT-qPCR did not match perfectly. These data confirmed the reliability of the RNA-Seq results. Briefly, as shown in Fig. 3 the RNA-Seq and RT-qPCR display consistent expression patterns for the examined genes. Specifically, the expression levels of all unigenes were higher in H. longipes roots than in other plant tissues, which is consistent with the RNA-Seq data. This indicates that the experimental results are reliable. Therefore, we conclude that RNA-Seq data presented here accurately represent differences in transcript expression between the alkamide-producer tissues (H. longipes roots) and the tissues that do non-produce these compounds (H. longipes leaves, H. annua leaves, and H. annua roots).

Phylogenetic analysis of the Ketosynthase domains among differentially expressed genes

In order to find genes that could be related to the biosynthesis of alkamides, the annotated transcriptomes were used to search for transcripts that coded for conserved domains of fatty acid synthases, but that also were shared with other related enzymes. We noticed that, among the list of differentially expressed transcripts, there were four annotated as coding for proteins containing the KS domain. This domain was chosen for further analyses not only because it is the most conserved domain in the FASs, but it is also present in other related enzymes like the PKS multi-enzymatic complexes and used for phylogenetic studies (You et al., 2014; Gallo, Ferrara & Perrone, 2013; Kohli et al., 2016). The KS domain catalyzes the condensation of two substrates depending on the type of enzyme, that will form a growing acyl chain (Brown, Slabas & Rafferty, 2009). We expected that enzymes containing this domain would participate in the biosynthesis of the acyl chain of alkamides. The phylogenetic analyses of eukaryotic FASs and PKSs using these conserved KS domains, has been used to emphasize the singularity of the enzymatic complexes, their evolutionary relationships, taxonomic distribution, and the level of conservation of these multi-modular enzymes (You et al., 2014; Gallo, Ferrara & Perrone, 2013; Kohli et al., 2016).

The list of annotated transcripts differentially and not differentially expressed in H. longipes and H. annua tissues was screened for transcripts coding proteins containing a KS domain. In order to better identify the type of KS domains coded by the different transcripts, we used the translated transcript sequences to perform a molecular phylogenetic analysis using the maximum likelihood method. The analysis included the KS domains of transcripts upregulated in H. longipes roots. As a comparison, also was included KS domains from H. longipes leaves and H. annua roots and leaves, which did not show differential expression but for which the annotation indicated that they coded proteins related to FAS or PKS type III of plants. These different KS domains were classified as: A, C, D, E or F for transcripts from H. annua (roots and leaves) and H. longipes (leaves), that were annotated as coding for a KS domain or KS domains related to FAS or type III PKS; and B for H. longipes roots upregulated transcripts. Other KS domains of FAS and PKS from mammals, microorganisms and other plants, obtained from NCBI listed in Table S1. KS domains encoded in transcripts of Echinacea purpurea were also included.

The results of the maximum likelihood phylogenetic analysis based on 89 KS sequences are presented in Fig. 4. As expected, the KS domains that were annotated as related to FAS from H. longipes D1–D11 and H. annua C1–C11, are grouped in a clade together with the FAS from Arabidopsis thaliana KS with a high bootstrap value (98%). This result suggests that these transcripts, obtained from H. longipes and H. annua, encode enzymes with function similar to the KS domain of A. thaliana, which belongs to β-ketoacyl-[acyl-carrier-protein] synthases III (EC 2.3.1.180), enzymes that catalyze the chemical reaction of acetyl-CoA + malonyl-[acyl carrier protein] and produces acetoacetyl-[acyl carrier protein] + CoA + CO2. These enzymes belong to the family of transferases, and particularly to those transferring groups other than aminoacyl groups. They participate in fatty acid biosynthesis and are found in the dissociated or type II FAS biosynthesis system that occurs in plants and bacteria. The KS domains related to type III PKS are represented by the domain found in chalcone synthases (CHS), the enzymatic complexes mediating the synthesis of flavonoids in plants (Abe & Morita, 2010). The non-differentially expressed KS domain sequences that were chosen due to their annotation as type III PKS from H. longipes (F1–F4) are grouped together with a high bootstrap value of 100%. However, they are grouped in a clade far apart from the KS of type III PKS from H. annua E1–E4, which cluster in a clade together with the KS of type III PKS from Arabidopsis thaliana with a high bootstrap value of 98%. Therefore, these transcripts labeled as E1–E4 may share a similar function to the type III from A. thaliana. In addition, the translated transcripts of H. annua A1–A4 are close to the clade of H. annua E1–E4, translated transcripts coding for type III PKS KS domains or Chalcone synthase (CHS). As for the FAS-related sequences, the transcripts coding for these domains are present in both species, which could suggest that they may not be involved in the biosynthesis of the acyl chain of alkamides.

Figure 4 Molecular phylogenetic analysis of the ketoacyl synthase (KS) domains protein sequences.

KS domain protein sequences differentially expressed in H. longipes and H. annua roots and leaves tissue were selected for phylogenetic analysis. KS domain protein sequences non-differentially expressed but annotated in the transcriptomes that coded for conserved domains of fatty acid synthases (FAS) and polyketide synthase (PKS) were used as comparison. The different KS domains selected were classified as group A, C, D, E or F for non-differentially expressed transcripts from H. annua (roots and leaves) and H. longipes (leaves). The group B for the upregulated transcripts in H. longipesroots. KS domains from: Echinacea purpura from Plant Biology Department, Michigan State University; and other KS domains of FAS and PKS from mammals, microorganisms and other plants, obtained from NCBI, were also were selected for phylogenetic analysis based on the Maximum Likelihood method. The numbers at nodes indicate bootstrap support, for 100 bootstrap replications. The analysis was conducted in MEGA 7.

The KS domains coded by the transcripts B1–B4 found to be upregulated in H. longipes roots, the species and tissue that produces alkamides, grouped with a high bootstrap value of 100% with KS (3-oxoacyl-[acyl-carrier-protein] synthase) of Echinacea purpurea roots. This group is exclusively formed by translated transcripts of E. purpurea and upregulated in H. longipes roots. This association suggest that the KS domains in both species probably have the same function and could be related to alkamide acyl chain biosynthesis, due to their functional nature. Close to this KS clade, KS domains from other plant species include Helianthus annus and Cynara cardunculus. This clade is not far from KS domains from the Chlamydiae phylum microorganism PKS specifically with the two orders: Chlamydiales and Parachlamydiales; and a single class Chlamydiales. These are obligate intracellular Gram-negative bacteria (Gupta et al., 2015).

In summary, the phylogenetic analysis revealed that the KS domains found in the annotated transcriptomes of the two species of the Heliopsis genus under study, are separated into at least four categories associated with plants: (1) The first KS domain, associated with plant FAS, includes general H. longipes, H. annua, and other plant transcripts; (2) A second domain type found exclusively in H. longipes root upregulated transcripts and E. purpurea. This KS domain, present in both species, is possibly related to the biosynthesis of the acyl chain of alkamides in H. longipes and E. purpurea. Interestingly, they were grouped in a clade far from FAS and PKS III KS reported in plants, but closer to microorganism PKS (Hertweck, 2009; Piel, 2010); (3) The third domain type present exclusively in H. longipes, associated with type III PKS is located far from the other KS domain of H. longipes, related to the alkamides biosynthesis. These H. longipes type III KS domain could be associated with other metabolites from H. longipes; (4) The fourth includes two clades of H. annua KS domains related to type III PKS.

Proposed model of a pathway for the biosynthesis of the acyl chain of alkamides

The acyl chain of affinin is a straight chain with an even number of carbons, which suggests that acetate units are involved in its biosynthesis as mediated by a FAS. The presence of double bonds in the chain at even positions may be justified by the deficiency of enoyl reductase (ER) enzymes in a modular manner. As observed in polyketides biosynthesis from bacteria, fungi and symbiotic marine microorganisms such as in sponges (Piel, 2009), where unsaturated bonds that are incorporated during the chain elongation. The differentially expressed genes encoding polyketide synthase domains and among the candidate transcripts, four transcripts encoding the AT, KS and DH, seven transcripts encoding the OR and eleven transcripts encoding the ACP enzymatic domains were found. Thus, a PKS pathway using all these enzymatic domains for the biosynthesis of the acyl chain moiety of H. longipes alkamides is proposed. The available information about the different domain activities in published PKS systems, allowed the assembly of a model of a biosynthetic pathway that could produce the C10-2E,6Z,8E acyl chain of affinin. This model incorporates the KS, AT, ACP, OR (including the keto reductase (KR) and ER), DH and TE domains previously found. The model proposes a four-module PKS system where the starting precursor molecule is acyl-CoA, and the extender unit is malonyl-ACP (Fig. 5).

Figure 5 Proposed affinin acyl chain biosynthesis mediated by polyketide synthase (PKS).

The proposed biosynthetic pathway exhibits a multi-modular architecture and requires acyl-CoA as the starter unit and malonyl-ACP as the extender unit. The KS domain catalyzes a decarboxylative Claisen (CO2) condensation between the growing chain and an extender unit attached to the ACP domain (gray circles), whereas the AT domain selects and loads an extender unit to the ACP. The oxide-reductase that includes the KR and DH domains sequentially reduces the β-keto group to a β-hydroxy group and α-double bond. Additional ER domains reduce the α-double bond to a saturated product, and TE cleaves the synthesized acyl chain. Enzymatic domains: ACP: acyl carrier protein, AT: acyl transferase, KS: ketoacyl synthase, KR: keto reductase, DH: dehydratase, ER: enoyl reductase, TE: thioesterase.

The proposed sequence of acyl chain biosynthesis, as evidenced by the differential expression supporting a four-modules multi-enzymatic complex, is as follows: in the first module (Aff-1), the AT (GO:0016740) domain accepts and loads the extender unit onto the ACP (GO:0016740) and produces malonyl-ACP. Then, the KS (GO:0016740) domain catalyzes a decarboxylative Claisen condensation with the acyl-CoA and the resulting product is a four carbon (C4) acyl chain with α- and β-keto groups. The KR, including OR (GO:0016491) and DH (GO:0016740) domains, sequentially reduce the β-keto group to a β-hydroxyl group and an α-E double bond, respectively. The α-unsaturated acyl chain is transferred by the AT (GO:0016740) to the second module (Aff-2) and is elongated by the ACP (GO:0016740) with another malonyl-ACP producing a C6-acyl chain. The same enzymatic reactions on the β-keto group as in the first module occur in module 2. The product is a C6-2Z,4E-dien-acyl chain, which is transferred to the module 3 (Aff-3) and condensed with malonyl-ACP to produce a C8-acyl chain with α- and β-keto groups. In this module, which consist of includes the KR including OR (GO:0016491) and DH (GO:0016740) domains, the β-keto group is reduced to a β-hydroxyl group and dehydrated to an α-E double bond. The ER (GO:0016491) enzymatic domain, present only in this module, reduces the double bond. In module 4 (Aff-4), the acyl chain C8-4Z,6E is once more condensed with malonyl-ACP to produce an C10-acyl chain with two keto α- and β- groups. A KR, including OR (GO:0016491) and DH (GO:0016740) domains, in this module reduces the β-keto group to a β-hydroxyl group, and the DH domains to an α-E double bond. The absence of an ER domain in this last module 4 results in the transference of the final C10-2E,6Z,8E-chain to the TE (GO:0016744) domain present at the end of this four module PKS. Table S3 includes the sequence and annotation of unigenes participating in the putative biosynthesis pathway and information about the alignments to Arabidopsis and other plant species.

The acyl chain length of the alkamides would depend on the number of modules present in the multi-enzymatic PKS complex, while the position and number of double bonds, would depend on the presence and activity of ER domains following the KR and DH domains, as reported for other PKS complexes (Hertweck, 2009). The combination of all these factors may then lead to the biosynthesis of acyl chains with different structures, which could explain the structural diversity of natural alkamides.

Discussion

The aim of this study was to obtain experimental information on global gene expression in roots and aerial tissues of two Heliopsis species and use this information to identify candidate genes involved in the biosynthesis of the acyl chain of alkamides, based on the fact that alkamides are differentially synthesized in some specific Heliopsis species and tissues. Alkamides accumulate in H. longipes roots, as observed in older plants contain higher levels of alkamides than roots of younger plants (García-Chávez, Ramírez-Chávez & Molina-Torres, 2004). These compounds are continuously being biosynthesized during plant growth, independently of external factors, suggesting that their biosynthetic genes are constitutively expressed in the tissues where alkamides are produced. RNA-Seq studies have facilitated the discovery of biosynthetic genes responsible for the production of plant-specialized metabolites in non-model plants, through the sequencing and comparison of transcriptomes of different tissues (Xiao et al., 2013). In the present work, the Illumina HiSeqTM 2500 technology was used to obtain the transcriptomes of roots and leaves of two Heliopsis species, where over 90% of the assembled unigenes matched with the genomic database of other plants. To the extent of our knowledge, this is the first report about global gene expression analyses of roots and leaves of Asteraceae species. The diversity of the GO terms related to the assembled unigenes, as demonstrated by functional GO assignments, revealed a variety of unigenes involved in specialized metabolite biosynthesis pathways.

Differential gene expression analysis allowed to identify genes that were highly expressed in H. longipes roots, but not in tissues that do not accumulate these metabolites: H. longipes leaves or H. annua roots and leaves. Because of their expression pattern, they are very likely related to the biosynthesis of alkamides. Among the annotated transcripts, we found some coding for the enzymatic domains KS, AT, ACP, DH, OR and TE, which presented higher expression in H. longipes roots than in leaves (Fig. 3A). The expression level of these genes was further evaluated by RT-qPCR and all unigenes tested showed higher expression in H. longipes roots (Fig. 3B). Given their preferential expression, which could explain why alkamides are produced and accumulate in H. longipes roots but not in H. longipes leaves nor H. annua tissues, these genes are potential candidates to code for the enzymes that perform the biosynthesis of the acyl chain moiety of alkamides.

To better understand the biosynthesis of the acyl chain of alkamides and considering the nature of the chemical reactions that could produce such a chain structure, we sought enzymatic domains in the translated differentially expressed transcripts that could perform such reactions. The first domain searched was the KS domain. This domain is very conserved in FAS, and as in other related enzymes, such as the fatty acid elongases (FAE) that act at the end of the FAS steps, or complexes such as PKS. It is responsible for the condensation of two substrates to generate or elongate an acyl chain, depending on where the enzyme that contains it is present. It can be assumed that such a domain would be participating in the initial steps of the biosynthesis of the acyl chain of alkamides. Indeed, transcripts encoding the KS domain were found among the differentially expressed set. The phylogenetic analysis (Fig. 4) revealed that the KS domains of the differentially expressed transcripts in H. longipes roots and the KS domain found in transcripts of E. purpurea, were grouped together with a high bootstrap value of 100%, suggesting that they share a common ancestor. This clade that included the KS domain of plants and tissues that produce alkamides can be called “Alkamides clade”. The KS domains in the alkamide clade are separated from the KS domains of FAS from H. longipes and H. annua leaves, as well as from the KS of the type III PKS from H. longipes and H. annua. Type III PKS is the chalcone synthase complex mediating the synthesis of flavonoids in plants (Abe & Morita, 2010).

Surprisingly, in the phylogenetic analysis comparing KS domains of FAS, PKS, and the putative alkamide acyl chain biosynthetic genes, KS domains of the latter were found in a clade close to PKS from microorganisms. Thus, PKS-related, specific genes, appear to be conserved within the alkamide producer species, suggesting duplications that led to the evolution of novel functions. A possibility is that these plants acquired these genes by horizontal transfer, as has been suggested for other PKS gene families (Kohli et al., 2016) or alternative splicing that can give rise to different transcript versions of a single gene. Previously, no KS domains resembling type I PKS KS domains have been reported in plants, possibly because of initial low sequencing depth. It would be very interesting to search carefully for similar transcripts in other plants, especially those known to synthesize alkamides.

Because of the clustering of KS domains coded by upregulated transcripts with KS domains found in type I PKS systems, found in certain microorganisms, we sought whether other similar domains present in those systems were also found coded by other upregulated transcripts. These results suggest alkamide synthesis is likely carried out by enzymes that are similar to type I PKS. These findings are important, as the differentiation of PKS and FAS will facilitate approaches investigating alkamides biosynthesis pathways in other plant species. They are also interesting as this type of systems were previously thought to be absent from plant species, and mostly present in microorganisms. One possibility is, of course, that these enzymes are expressed by endophytes that are exclusively present in the root tissues of H. longipes. However, since RNA-Seq analysis was performed using polyadenylated RNA, it is unlikely that prokaryotic RNA was sequenced, though fungal transcripts would not have been filtered out. Nevertheless, we would expect fungal transcripts to be present at much lower levels than plant transcripts. Moreover, putative endophytes that are exclusively present in H. longipes roots have not been observed in the in vitro root culture in the lab to date.

To propose the novel alkamides biosynthetic pathway, we considered the chemical structure of the affinin acyl chain, and the characteristics of the known products of FAS or PKS. Fatty acids derived from FASs comprise completely reduced acyl chains, and the polyketides derived from PKSs include partially unsaturated acyl chains. In the polyunsaturated fatty acids (PUFAs), the double bonds are incorporated by fatty acid desaturases (FADs), iron-dependent metalloenzymes, on the fatty acids released from the FAS enzymatic complex (Gagné et al., 2009; Shanklin et al., 2009). In contrast, the double bonds in the polyketides are the result of the enzymatic activity of KR, DH, and ER domains (Hertweck, 2009) during the chain elongation.

The pathway proposes that the candidate enzymes are involved in the biosynthesis of the affinin acyl chain in H. longipes roots through a pathway that is similar to the polyketide biosynthesis pathway. This model is further referred as “PKS alk”. The proposed PKS alk (Fig. 5), would incorporate and keep double bonds during the elongation of the chain. In order to accomplish this, it would contain four modules, which could be named: Aff-1, Aff-2, Aff-3, and Aff-4, where each incorporates two carbons. It would require acyl-coenzyme A (CoA) as the starter and malonyl-acyl carrier protein (ACP) as the extender unit. The C10-2E,6Z,8E as a polyunsaturated acyl chain, would also require PKS alk to have a partial reduction activity to produce the molecular structure of the acyl chain of alkamides. The selected set of genes contain the domains required to support these reactions, although further studies are required to confirm their participation in the synthesis of the acyl chain of alkamides. The proposed system (PKS alk) would work in a modular fashion as in the known type I PKSs (Hertweck, 2009). This multienzymatic mechanism is detailed in Fig. 5. In order to obtain the affinin acyl chain specific structure, the KR and DH domains would perform a partial reduction in modules 1 (Aff-1), 2 (Aff-2), and 4 (Aff-4), which are referred to as γ-modules because they contain only KR and DH domains. In comparison, a full keto group reduction would be provided by module 3 (Aff-3), which contains ER in addition to KR and DH, and is referred to as the δ-module in agreement with the nomenclature recently suggested by Keatinge-Clay (2017). In this pathway, the AT substrate would be malonyl-CoA, as observed in FASs and some other PKSs (Surup et al., 2014). After the first two biosynthetic modules perform their activities, the conjugated double bonds in 4-trans and 2-cis positions would be introduced, as the DH in the first module acts on a D-β-hydroxy-acyl intermediate to form a trans double bond in the Aff-1(γ) module. However, in the Aff-2(γ) module, the DH would generate a cis double bond during the dehydration of an L-β-hydroxy intermediate. This mechanism, by means of which the DH generates cis double bonds when a trans double bound is inserted in the previous module during the synthesis of an acyl chain, has been observed in several PKS type I enzymes in microorganisms (Kalan et al., 2013). Once the alkamide acyl chain is synthesized, the offloading machinery TE in the Aff-4(γ) module, would be responsible for freeing the α-unsaturated acyl chain from the PKS complex. This TE should differ from the FASs TE. It would accept an α-unsaturated substrate and, by acting as a logic gate, would determine the substrate’s fate (Horsman, Hari & Boddy, 2016). In the last step of the proposed pathway, the TE would act specifically on the acyl α-unsaturated chain, such as the affinin acyl chain C10-2E,6Z,8E. Notably, this TE should be similar in all of the offloading alkamide PKSs in order to maintain the characteristic α-unsaturation of these structures. In addition, the TE may be involved in the determination of the amine to which the acyl chain will be linked via an amide bond, as the final step of alkamide biosynthesis. The TE of FASs and PKSs catalyze substrate offloading from ACPs. Acyl chains bound to the arm of ACPs are loaded onto the active site serine of a TE and then released via a nucleophilic attack (Horsman, Hari & Boddy, 2016). Regarding alkamide biosynthesis this enzyme should be specific, as it would transfer the chain to an amide and not to an ester, and should support the obligated presence of an α-unsaturated acyl chain as defined for alkamides. More detailed studies would be required to explain the presence of the bornyl ester of C10-2E,6Z,8E reported for H. longipes roots only (Molina-Torres et al., 1995).

The “PKS alk” model can be used as a basis to design experiments to demonstrate it is indeed responsible for alkamide biosynthesis, and to better understand it. According to this mechanism, there would be a great similarity between PKS alk and plastid type I FAS biosynthetic pathway. Both are giant multienzymes complexes catalyzing all steps of the biosynthesis from acetyl- and malonyl-CoA precursor extension and utilizing the KR, DH, and ER, to reduce the keto group. However, the main difference between the PKS alk and Type I FAS would be that the FAS type I is constituted by seven modules, while the proposed PKS alk, contains four putative modules. In modules one, two and four of the PKS alk the enzymatic domain with ER activity would be absent, while in FAS, all modules are similar and contain the ER domain. Besides, while the FAS is present in all plants, the alkamides have been reported only in specific plant species of different genus and families, suggesting that the PKS alk also presents a limited distribution among the plant species. Although several plant species of taxonomically different families produce alkamides such as the Asteraceae, Piperaceae, and Rutaceae, the PKS alk pathway, proposed in this study, maybe also mediating the biosynthesis of all the alkamides in those species phylogenetically separated for example, Zhantoxylum spp (Rutaceae) and Ctenium aromaticum (Poaceae).

Conclusions

In the present study, the first transcriptomic high-throughput sequencing of the Mexican endemic plants H. longipes and H. annua is described. The comparative transcriptome analysis performed for the two Heliopsis species led to the selection of a set of candidate genes putatively involved in the affinin acyl chain biosynthesis, allowing to propose a model for alkamide acyl chain biosynthesis by a PKS complex, here named PKS alk. This complex would be partially reductive and related to type I PKSs, and present in plants that produce alkamides. These findings provide a framework for molecular and biochemical studies to confirm the activity of the proposed pathway. Further studies can also help to better understand the underlying regulatory mechanisms involved and the physiological role of these multifunctional molecules in plants.

Supplemental Information

Supplemental Information 1 Accession numbers and organism of origin of the sequences used for the phylogenetic analysis of KS domains.

Click here for additional data file.

Supplemental Information 2 Primer pairs used to detect the expression levels of candidate genes by quantitative real-time PCR.

Click here for additional data file.

Supplemental Information 3 Differentially expressed Candidate genes participating in the putative biosynthesis pathway of the acyl chain of affinin (PKS alk model).

The table includes the Enzymes name, Unigenes, Sequences and GO that presented increased accumulation in H. longipes roots and a relevant putative function according to the annotation, and the Locus, % identity, e-value, score and information about the alignments to Arabidopsis (green section) and other plant species (blue section) is presented, for the proposed multi-enzymatic complex PKS alk for the synthesis of Affinin.

Click here for additional data file.

Supplemental Information 4 qRT-PCT raw data.

This data was used to obtain Figure 5: Relative expression of nine candidate genes involved in the putative alkamide acyl-chain biosynthetic pathway.

Click here for additional data file.

We thank Enrique Ibarra-Laclette, Juan Carlos Ochoa Sánchez for their technical support in this project, David Ramírez-Noya for the collection and taxonomic identification of the Heliopsis species, Elias Nieto Resendiz for providing Heliopsis longipes tissues, and Laila Partida-Martinez, Francisco Barona Gómez, Luis José Delaye, and Carolyne Smith, a Peace Corps Response volunteer, for reading the manuscript.

Abbreviations

ACP acyl carrier protein

AT acyl transferase

DH dehydratase

ER enoyl reductase

FAS fatty acids synthetase

GO gene ontology

KR keto reductase

KS ketosynthase

OR oxide reductase

PKS Polyketide synthase

RNA-Seq RNA sequencing

RT-qPCR quantitative real time polymer chain reaction

TE thioesterase

TF transcription factor

Additional Information and Declarations

Competing Interests

Author Contributions

Data Availability

The authors declare that they have no competing interests.

Génesis V. Buitimea-Cantúa conceived and designed the experiments, performed the experiments, analyzed the data, prepared figures and/or tables, authored or reviewed drafts of the paper, and approved the final draft.

Nayelli Marsch-Martinez conceived and designed the experiments, performed the experiments, analyzed the data, prepared figures and/or tables, authored or reviewed drafts of the paper, and approved the final draft.

Patricia Ríos-Chavez performed the experiments, authored or reviewed drafts of the paper, and approved the final draft.

Alfonso Méndez-Bravo analyzed the data, authored or reviewed drafts of the paper, and approved the final draft.

Jorge Molina-Torres conceived and designed the experiments, performed the experiments, analyzed the data, prepared figures and/or tables, authored or reviewed drafts of the paper, and approved the final draft.

The following information was supplied regarding data availability:

Sequences are available at NCBI BioProject: PRJNA433759.

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
