# Peer review of "Global gene expression analyses of the alkamide-producing plant Heliopsis longipes supports a polyketide synthase-mediated biosynthesis pathway"

_PeerJ, doi:10.7717/peerj.10074_

## Round 0.1 · original submission · Major Revisions

Dear authors,

As you can see our reviewers recommend a substantial revision. Make sure that you make an adequate revision, as these reviewers will check your revision.

Kind regards

Michael Wink
AE

Reviewer 1 ·

Basic reporting

No comment

Experimental design

No comment

Validity of the findings

No comment

Additional comments

Comments:

Manuscript # peerj-45349

The manuscript describes the identification of candidate genes for alkamide biosynthetic pathway. The authors performed RNA-seq on plant materials with tissues from alkamide-producing plant Heliopsis longipes and its related species H. annua, a non-alkamide-producing plant. The sequencing data were then assembled, annotated and compared. Based on the differential expression data, ketosynthase domain search, phylogenetic analyses and RT-qPCR data, the authors proposed a model for alkamide acyl chain biosynthesis, a PKS type I-like complex (PKS alk) with four modules which had not been reported previously for plant kingdom.

Overall, the experiment design was permissible and apparently well-executed, and the manuscript is well written. I have relatively minor points for the authors to address.

If the journal allows, consider adding a small section for abbreviations. It seems difficult to read it through without it.

The figure titles and legends are missing.

Consider using “RT-qPCR” in your manuscript by following the MIQE guidelines (Bustin et.al. 2009, Clinical Chem 55:4). Of course, it is up to the editor and the journal standard.

Fig 5 – how consistent are your RT-qPCR results with the RNA-seq data? Is it possible to include the comparison of the “target genes” with the sequencing data in the figure?

Reviewer 2 ·

Basic reporting

This manuscript aims to address an interesting and important challenge - the biosynthetic route to alkamides in plants. The overall strategy is clear - comparative transcriptomics of two related Heliopsis species, one that is reported to produce alkamides and one that does not. The producing species, H. longipes, is reported to produce these compounds in the roots but not the leaves, so enabling comparisons to be made both at the levels of species and tissues. The writing needs some improvement and is ambiguous in places. While the generation of transcriptome resources is valuable, there is far too much unsupported speculation about the types of enzymes that may be involved. Functional analysis is needed in order to back this up.

Experimental design

The two species used in these experiments were harvested at different times of the year, and so seasonal differences may influence the results. A more rigorous approach would be to use material grown under controlled conditions (in a controlled environment glass house or growth chamber). It would also have been reassuring to analyse the alkamide content of the material used for transcriptomics to confirm the expected differences between species and organ types.

Validity of the findings

The apparent finding of type I PKS enzymes in plants is surprising and warrants a good degree of caution. A more detailed analysis is needed, both in terms of phylogenetic comparisons and text of function. It would be very interesting if such enzymes are involved in alkamide biosynthesis in the roots of H. longipes. The possibility remains that such sequences could originate from microbial symbionts or endophytes that grown in/on the roots. Without further investigation this work is very preliminary in nature.

Additional comments

This is a really interesting system. I encourage the authors to carry out further investigation to test/substantiate their hypotheses.

---

## Round 0.2 · Minor Revisions

Dear authors.

Under further review of the manuscript structure it appears that there are some connections to data that are not resolved. I am generally onboard with the manuscript presentation, and applaud the addition of annotations to the transcriptome data analyzed; however, there is no connection whatsoever to any of the sequences discussed. There is a pointer to the raw data-sets; however, the ability to connect such sequence unigenes to the alluded to annotations is not evident.

Likewise, the form in which these annotations should be presented should connect the annotations alongside the appropriate numerical GO (e.g. GO:12345) terms that match with the text annotations. I would assign this as requiring moderate revision until properly available unigene sequences can be presented with a mechanism to connect the GO: annotations to the assigned unigenes.

If data will not be provided for the successfully annotated 29,316 unigenes in H. longipes and 29,870 unigenes in H. annua then at least highlighted sequences important to the biosynthesis pathways and formerly discussed should be included. These may be added as supplemental files, but none of the requested appeared there.

I will leave this for you to adjust before reviewing the next revision. It's close but does need to be more formal in its presentation. We look forward to the revision.

Reviewer 3 ·

Basic reporting

No comment (some minor language errors, for example "in any the other" in line 47)

Experimental design

No comment

Validity of the findings

No comment

Additional comments

I think this work deserves publishing: it gives at least an alternative explanation for the biosynthesis of Nalkylamides in plants. As such, and while it is not a complete proof awaiting additional experiments, it can be further challenged by other investigations.

---

## Round 0.3 · accepted · Accept

Thank you for adding the suggested edits. It now reads in a fashion which will lend itself to validation and help to open discussions to move the work forward. I approve of the manuscript form and recommend it to move forward toward publication. Congratulations on your efforts.